# Diagnostic and Treatment of Spinal Fracture and Luxation in Italian Wolves (*Canis lupus italicus*)

**DOI:** 10.3390/ani12213044

**Published:** 2022-11-05

**Authors:** Domenico Fugazzotto, Chiara Costa Devoti, Maria Pia Dumas, Chiara Teani, Elisa Berti, Offer Zeira

**Affiliations:** 1Ospedale Veterinario San Michele, Via I Maggio 37/A, 26838 Tavazzano con Villavesco, 26900 Lodi, Italy; 2Centro Tutela e Ricerca Fauna Esotica e Selvatica-Monte Adone, 40037 Sasso Marconi, Italy

**Keywords:** wolf, dog, spinal fracture, luxation, spine stabilization

## Abstract

**Simple Summary:**

This paper describes the clinical presentation, neurological examination, diagnostic findings and treatment of spinal fractures and luxations (SFLs) in Italian wolves *(Canis lupus italicus)*. Given the lack of literature on spinal pathologies in this wild species, our clinical choices were based on current medical literature on dogs.

**Abstract:**

The medical records of 14 Italian wolves (*Canis lupus italicus*) with a vertebral fracture or luxation (SFL) between C1 and L7 treated at Ospedale Veterinario San Michele from 2017 and 2022 were reviewed. The most common cause of SFL was “road traffic accident”. Neurological signs were graded from 0 to 6 using a modified Frankel scale. Spinal fractures occurred in C1–C5 in 1 case, in T3–L3 in 11 cases and in L4–L7 in 2 cases. Six wolves were euthanized without treatment because they presented paraplegia without deep pain perception (DPP). Two animals with motor function were treated conservatively, and later on one of them was euthanized because of neurological impairment. Six wolves were surgically treated. Seven wolves had good neurological recovery, and six of them were released into the wild. Our results suggest that wolves with DPP before surgery may have a good functional recovery.

## 1. Introduction

Spinal fractures and luxations (SFLs) in small animals are commonly associated with severe external trauma and occur in approximately 6% of cases involving neurologic deficits indicative of spinal cord dysfunction [1,2]. In dogs, SFL are commonly caused by road traffic accidents (RTAs) and falls from heights [3,4,5]. Other causes of SFL include trauma due to animal attacks, gunshot wounds and pathologies such as neoplasia, infections and metabolic diseases [6]. The thoracolumbar tract statistically represents the most common region of the rachis affected by SFL [7]. Therapeutic management can be either conservative or surgical. Conservative treatment involves external immobilization with splints and bandages, cage confinement, exercise restriction, analgesia and anti-inflammatory medications [8]. Surgical treatment involves spinal stabilization with pins or screws and polymethylmethacrylate (PMMA), vertebral body plating, spinal stapling or external skeletal fixation [9,10]. This study reviews the medical records of 14 Italian wolves (*Canis lupus italicus*) with SFL and describes their neurological status, SFL location, treatments and outcomes.

## 2. Materials and Methods

### 2.1. Ethical Statement

According to the Italian law 152/1992, the Wildlife Protection Centers (C.R.A.S) can contact veterinary facilities involved in the care and welfare of wild animals. Centro Tutela e Ricerca Fauna Esotica e Selvatica—Monte Adone, a renowned wildlife rehabilitation center and a national point of reference for wildlife management, with more than 10 years of experience in wolf rescue, treatment and rehabilitation and with dedicated personnel and facilities, referred patients to the San Michele Veterinary Hospital (SMVH). Italian wolves are classified as a vulnerable species on the red list by the IUCN in Italy and are considered at high risk of unnatural (human-caused) extinction without further human intervention [11]. Several projects are underway in Italy to preserve and restore the wolf population. In view of their clinical presentation, rescued wolves were referred to an ECVIM board eligible veterinary neurologist for specialized evaluation, diagnostics and treatment. Wolves were released into the wild according to art. 26 comma 6 bis and 62 comma 1 regional law 8/94. Euthanasia was performed according to international guidelines and complied with the relevant national legislation.

### 2.2. Clinical Examination

Fourteen Italian wolves were referred to the SMVH for neurological evaluation between 2017 and 2022. Medical records were reviewed, and data regarding the cause of the SFL, clinical examination, comprehensive blood works (hematology, biochemistry and blood-gas analysis), abdominal/thoracic ultrasound, radiological examination, computed tomography (CT) and magnetic resonance imaging (MRI) were collected. For the purpose of this study, the reported data cover wolves’ age, sex weight, the cause of SFL and neurological status. The cause of the SFL was as described by the person who initially found the wolves and contacted the rescue center. All patients were referred to our hospital after initial medical stabilization (fluid therapy, analgesia and limb immobilization in cases of fractures). At the SMVH, neurological examination was performed on conscious animals using muzzles, blindfolds and gentle restraint by operators experienced in wildlife management. Neurological status was graded from 0 to 6 using the modified Frankel scale as follows: grade 0, loss of DPP; grade 1, no purposeful movement with deep pain perception; grade 2, no purposeful movement with DPP; grade 3, non-ambulatory but able to perform purposeful movement; grade 4, proprioceptive deficits and/or ataxia; grade 5, hyperesthesia without neurological deficits; grade 6, normal neurological status [12]. As in dogs, DPP was assessed witnessing a behavioral response after pinching digits of all four limbs and tail with a hemostat forceps. All animals received an intermuscular injection of tramadol (3 mg/kg IM, Altadol, Formevet, Milan 20144, Italy) at the rescue center a few hours before arriving at SMVH and an intramuscular injection of methadone (0.2 mg/kg; Semfortan, Eurovet Animal Health B.V, Handelsweg, 5531 AE Bladel, Netherlands) after the neurological examination. The more invasive workup exams (blood work, radiograph, CT and/or MRI scans) were performed after sedation or anesthesia in order to minimize the wild animals’ stress, ensure operator safety and provide data of the best possible quality for diagnosis and surgical planning. 

### 2.3. Diagnostic Procedures

In all cases, the diagnosis of SFL was based on radiography, MRI, CT or a combination of these modalities. Radiograph examinations were performed using an analog radiographic table with a digital radiographic system (Foschi X1 PLUS with ARIA® Software, Demas SRL, Rome, Italy). Thoracic, abdominal and spinal radiography was performed in sedated animals to perform an initial evaluation of the suspected spinal injury and of other possible trauma to other body systems (e.g., pneumothorax, limb fractures). If necessary, abdominal and/or thoracic ultrasound was performed (point of care abdominal, thoracic and cardiac ultrasound) to better assess patients before prolonged general anesthesia and mechanical ventilation for imaging and surgery. Abdominal ultrasound was performed using an Esaote My Lab 40 ultrasound machine (Esaote S.p. A, Genova, Italy) with micro-convex and linear probes. MRI and CT studies were performed using low-field veterinary MRI (Vet Grande-MR, Esaote, Genova, Italy) and a Siemens CT scan (Siemens Somatom go.Now, Germany) (Figure 1). CT scans were performed to assess the exact location of the injury and characteristics of the SFL, to identify patient-specific anatomical landmarks and evaluate soft tissue trauma. MRI scans were performed to evaluate possible spinal cord injuries such as compressions, contusions, hemorrhages or lacerations. Three-dimensional imaging techniques were combined when CT scan results suggested a possible spinal cord laceration, hemorrhage or compression.

### 2.4. Treatment and Follow-Up

Compassionate euthanasia or treatment (conservative or surgical) was performed based on the neurological status and type of SFL. The surgical techniques were chosen according to the type and location of the SFL and the surgeon’s preferences. All patients were premedicated with dexmedetomidine (4 μg/kg i.m.; Dextroquillan, FATRO s.p.a., Ozzano dell’Emilia BO 40064, Italy), ketamine (1 mg/kg i.m.; Lobotor, ACME SRL, 42025, Cavriago RE, Italy) and methadone (0.2 mg/kg i.m.). General anesthesia was induced with propofol (1–4 mg/kg, Proposure, Boehringer Ingelheim Animal Health Italia s.p.a, 20139, Milano, Italy) to achieve orotracheal intubation and was maintained with isoflurane (IsoFlo, Zoetis Italia SRL, 20124, Milano, Italy) in oxygen with mechanical ventilation. The wolves were monitored intraoperatively with continuous ECG, pulse oximetry, non-invasive blood pressure, capnography and halogenates. Intraoperative analgesia was maintained with continuous fentanyl (3–6 μg/kg/h; Eurovet Animal Health B.V, Handelsweg, 5531 AE Bladel, Netherlands) infusions according to the patient’s pain level. Radiographic images were obtained immediately after surgery to evaluate implant positioning. Clinical outcomes were evaluated using serial clinical follow-up examinations performed by a veterinary surgeon at the wildlife rehabilitation center housing the wolves during recovery. Patients were classified as neurologically normal (grade 6), improved with residual dysfunction (higher neurological grade after treatment), unchanged (same neurological grade after treatment) or worsened (lower neurological grade after treatment). Follow-ups lasted from 3 to 16 weeks, except for one patient alive and housed at the rescue center at present. Follow-up was interrupted at the time of release into the wild for 6 patients and at the time of his euthanasia for one patient. 

## 3. Results

### 3.1. Case Analysis

Our study group included 14 Italian wolves (nine males (60.3%) and five females (39.7%)), aged 7 to 50 months. The median body weight was 22.4 (8.0–34.0) kg. RTA was the cause of SFL in 12 of 14 cases (85.8%). The cause of the trauma was unknown in one case, and a vertebral subluxation was due to severe discospondylitis in one wolf. Neurological status was grade 0 in six patients (49.9%), grade 4 in three patients (21.4%) and grade 3 in five patients (35.7%) (Table 1). None of the animals had clinical signs of head trauma.

### 3.2. Imaging

SFLs were confirmed using a digital radiography system and three-dimensional imaging techniques such as MRI and CT. All 14 animals underwent latero-lateral and dorso-ventral radiographic projections of the spine, thorax and abdomen under sedation. Thirteen wolves (92.8%) underwent a total body CT examination, and 11 wolves (78.5%) underwent MRI of the affected spinal segment (Figure 2). Thirteen wolves (92.9%) presented a combination of fractures and luxation/subluxation, and one wolf (7.1%) had only a T12-T13 subluxation secondary to severe discospondylitis, with no fractures. Four wolves (28.6%) presented fractures of two or more vertebrae. Spinal fractures were localized in C1–C5 in one wolf (7.1%), T3–L3 in 11 cases (78.6%) and L4–L7 in two patients (14.3%). Fractures were considered unstable in 12 of 13 fractured wolves (92.3%) because of the involvement of more than one vertebral compartment. At MRI, no wolves showed signs compatible with traumatic disk herniation, six animals (42.9%) presented extended intramedullary hemorrhage and two wolves (14.3%) had a laceration of the spinal cord. Concurrent orthopedic injuries were observed in six patients (42.9%) and involved fracture of the pelvis (n = 3), long bones (n = 2) and ribs (n = 1).

### 3.3. Treatments

The six paraplegic wolves (42.9%) without DPP (grade 0) were euthanized. Two wolves (14.2%) were treated conservatively. Of these, one had neurological grade 4 due to a T12–T13 subluxation secondary to severe discospondylitis. The second patient, with a neurological grade 3, had severe scoliosis due to displacement of partially healed fractures of the T5, T6 and T7 vertebrae. The first wolf was conservatively treated with a body-splint and a 10-week oral course of clindamycin 20 mg/kg s.i.d. (Antirobe, Pfizer Italy, Latina 04100, Italy) and marbofloxacin 2 mg/kg s.i.d. (Marbocyl P, Vetoquinol Italia, Bertinoro 47032, Italy) until resolution of neurological signs [13]. The latter was treated with a body-splint. 

Six wolves (42.9%) were treated surgically. Preoperative neurological status was grade 3 in four wolves and grade 4 in two wolves. Two (33.4%) of six cases were stabilized using screws and polymethylmethacrylate (PMMA) implants, while the remaining cases (66.6%) were stabilized using dorsal spinal stapling. *For the* spinal stapling, a Steinmann pin was contoured to act as a staple around spinous processes spanning to the site of injury. The size of the Steinmann pin was variable according to the size of the patient. Where possible, three vertebrae cranial and caudal to the injury were included in the staple. The fixation of the pin was achieved by drilling small holes through the base of *the* spinous processes. Loops of cerclage were then threaded in each hole and tightened around the pin. For the wolf with *an* L3 fracture, 3.5 mm cortical screws were used, and the angle of insertion and reference points were *chosen* according to *the* CT results. Fixation was performed unilaterally, using two screws on the fractured vertebra, two on the cranial one and two on the caudal one. Reduction forceps were used to maintain the unstable intervertebral articulation in a proper position while PMMA was applied and hardened. During curing of *the* PMMA, the surgical site was flushed with sterile 0.9% saline to decrease thermal injury to the surrounding soft tissues. Only one *wolf* presented with a cervical fracture of the vertebral body of C2, which was treated with 2 mm cortical screws (two transarticular screws between C1 and C2, two screws on the transverse body of C2) all encased in a reinforcement of PMMA. None of the patients required surgical access to the spinal cord for spinal decompression during spinal fixation. Four of the wolves who underwent SFL stabilization had concurrent orthopedic injuries. These fractures were surgically addressed during a second anesthesia because spinal stabilization was prioritized as an emergency treatment. Post-operative care and analgesia were adapted according to the neurological and clinical status of each patient. For long-term analgesia, tramadol (4 mg/kg t.i.d.) was administered orally for 10 days. In addition, all wolves received antibiotic therapy with oralcefadroxil (Cefa-Cure Tabs, MSD, Segrate 20090, Italy) (20 mg/kg orally s.i.d.) for 10 days and an anti-inflammatory course of prednisolone (Prednicortone, Dechra, Zuiveringweg 8243, Holland) starting from 0.5 mg/kg s.i.d. for the first 7 days and tapering gradually over the next 7 days. 

### 3.4. Follow-Up

Seven wolves showed neurological improvement after therapy. Five animals treated surgically achieved complete neurological recovery (grade 6), and one improved showing only residual spinal ataxia (grade 4). None of the patients who underwent surgical treatment showed worsening of their neurological status. No vertebral implants showed signs of complications, and five of six wolves surgically treated were released back into the wild. The wolf with residual ataxia is currently alive and housed in the rehabilitation center. The wolf conservatively treated with a body-splint was euthanized 3 weeks after diagnosis because of worsening of neurological status with progressive loss of sensorimotor function. The only surviving wolf who received conservative treatment was released back into the wild when neurologically normal.

## 4. Discussion

As in dogs [3,4,5], the most frequent cause of SFL in our population was RTA. Only one wolf had a sub-luxation secondary to an infectious process (discospondylitis). The diagnostic findings of SFL were confirmed by radiographs combined with three-dimensional imaging techniques (MRI and CT scan). One of the most important aims of diagnostic imaging in spinal trauma is to detect column instability. Most radiologists use the three-compartment model to identify vertebral instability. This method divides osseous and structural soft tissues of the column into ventral, middle and dorsal compartments. Damage to at least two compartments predicts vertebral instability and requires internal or external fixation [14,15]. Radiography provides an inexpensive and rapid means for initial evaluation of the vertebral column following trauma but has several limitations [16]. First, this technique heavily relies on accurate positioning of immobilized patients, and immobilization may require sedation. Nonetheless, after sedation, muscle relaxation and patient manipulation can potentially lead to iatrogenic spinal cord trauma. Loss of voluntary paraspinal contraction can increase the risk of subluxation in unstable vertebral segments. Second, radiography has only moderate sensitivity for fractures (72%) and subluxations (77.5%) and does not reliably rule out potentially unstable vertebral column lesions [17]. Radiography has been performed in our cohort to initially evaluate the spinal lesions and to identify any other signs of thoracic or abdominal trauma that can influence patient prognosis and immediate survival during prolonged anesthesia and surgery. Due to the temperament of the wild animals, all our patients were sedated before radiography. Cross-sectional imaging techniques, such as CT and MRI, overcome many limitations of survey radiography. Their main advantages include accuracy, multiplanar evaluation and reduced need for extensive patient manipulation [18,19]. CT can be considered the gold standard for assessing the osseous component of the vertebral column, because of its high sensitivity (up to 100%) for detection of acute osseous lesions, such as SFL [20]. Moreover, CT is commonly used in clinical research because the intra-observer agreement for fracture detection and classification is reported to be substantial to near perfect [21]. CT was performed in our population to evaluate the exact location, characteristics and morphology of the SFL and to allow familiarization with the anatomy around the injury. Three-dimensional reconstruction from CT images provides additional anatomical information on bone contours for appropriate surgical planning, including positioning, surgical approach and implant selection. In agreement with previous literature [15], fractures in our population were more common in the thoracolumbar spine. Advanced diagnostics found that most fractures of our wolves were unstable and required surgical stabilization, except in two cases, treated conservatively. In our study, MRI was performed when CT scan results suggested intramedullary or compressive spinal cord lesions and in order to give more accurate prognostic factors. MRI is the modality of choice for diagnosis of spinal cord injuries because of its ability to assesses intervertebral discs and ligaments and to detect vascular or soft tissue injuries. In addition, MRI can differentiate between spinal cord hemorrhage and oedema, and it is able to detect acute intervertebral disk herniation or disruption of the spinal cord [19]. There is mounting evidence that MRI can detect the presence of fractured vertebrae aided by associated soft tissue changes such as ruptured ligaments or changes in epaxial muscles [15]. However, MRI cannot reliably replace CT for detecting SFL. Gallastegui et al. (2018) reported that complete agreement between MR and CT results for exact fracture location is achieved only in 14.3–32.6% of fractured vertebra. According to Gallastegui’s results, MRI can miss up to 79% of fractures in some vertebral compartments. These data suggest that even if MRI can be a reasonable substitute when CT is unavailable, clinicians should opt for CT imaging to assess spinal osseous structures for evidence of trauma and fracture morphology [22]. In our patient sample, MRI showed paraspinal muscle injuries, osseous fractures or luxations, spinal cord compression and spinal cord swelling, as seen in dogs [15]. In wolves with neurological grade 0, MRI showed vertebral dislocation and consequent extensive spinal cord injury, with the presence of intramedullary hemorrhage and in some cases complete rupture of the cord. Nevertheless, in patients who underwent surgery, MRI showed no need for surgical access to the spinal cord to address compression caused by extradural hematomas, traumatic disc herniation or bone fragment displacement. The authors are aware that a combination of CT and MR is not a cost-effective work-up and cannot be afforded by everyone. We combined these techniques in order to be as accurate as possible in diagnosis, prognosis and surgical planning in wildlife animals, which are difficult to manipulate and have a limited timeframe for hospitalization. Six wolves (42.9%) were euthanized because of their poor neurological status (grade 0) and the lack of literature that could support a possible improvement of their clinical conditions. The search for tools to predict early and long-term postoperative motor function recovery in paraplegic patients after an acute spinal trauma is crucial in human and veterinary medicine. In particular, early predictors of motor function may expedite selection of patients with unfavorable prognosis for early implementation of novel therapies [23]. Wang et al. (2017) reported that the presence of DPP before surgical decompression in paraplegic dogs had a sensitivity of 73.3% and specificity of 75% in predicting early recovery of motor function and performed better than quantitative MRI. As for long-term prognosis, the presence of DPP in non-ambulatory dogs with thoracolumbar intervertebral disk disease has been associated with positive outcomes in nearly 100% of patients, and the absence of DPP has been correlated with a recovery rate of approximately 50%. Moreover, a small percentage of dogs without DPP may regain their motor function until 6 months after surgery, regardless of the recovery of deep nociception [24]. In addition, loss of DPP associated with extensive spinal cord injury can also predispose dogs to ascending or descending myelomalacia, the former being a fatal and irreversible medical condition [25]. These considerations are useful during communication between veterinarians and dog owners. Owners of paraplegic animals without DPP must acknowledge the high probability that their pet may never recover motor function and may have a permanent disability. In this respect, the management of paraplegic wolves in captivity would present medical and ethical challenges as well as not being in line with current regulations on wild animal welfare. Our decision to perform compassionate euthanasia was supported by absence of DPP associated with extensive damage or complete rupture of the spinal cord detected on MRI. The most commonly used surgical technique for vertebral stabilization in the present study was spinal stapling (four wolves), and neurological status improved in all treated wolves. Before surgery, two of four wolves were non-ambulatory but able to perform purposeful movements, while two animals had only spinal ataxia and proprioceptive deficits. The remaining two wolves were treated with screw–PMMA implants. The first case presented a vertebral body fracture at C2, and the second presented a vertebral body fracture at L3. Both wolves were non-ambulatory but able to perform purposeful movements before surgery. Several techniques have been used for stabilization of SFL in the thoracolumbar vertebrae in cats and dogs, each with distinct advantages and disadvantages regarding invasiveness and ease of application, fixation stability, clinical outcomes and applicability to various portions of the spinal column and to patients of different sizes. These techniques include external immobilization, spinal stapling, spinous process plating, vertebral body plating and spinal process plating combined with application of Kirschner–Ehmer apparatus, composite fixation with pins or screws and PMMA implants and stabilization with external fixators [3,4,5,6,7,8,9,10]. However, few studies have assessed outcomes of specific techniques in a large number of animals [3,10,26,27,28]. Spinal stapling involves applying parallel stainless-steel pins through and along the dorsal spinous processes on either side of the fracture, with additional wire stabilization through the dorsal spinous processes. Its advantages include reduced invasiveness, resistance to ventral bending [28,29], surgical ease, limited soft tissue dissection and the ability to perform concurrent hemilaminectomies [28]. The biomechanical advantage of these dorsally applied fixation devices is the resistance to ventral bending forces associated with thoracolumbar SFL because the implant is positioned on the tension side of the vertebral column, limiting flexion or extension [30]. In contrast, spinal stapling is less successful in patients with rotationally unstable lumbar SFL or in heavy patients with substantial bending and rotational instability [28]. According to a limited number of studies in small animals, spinal stapling has been suggested for SFL in cats and dogs weighing less than 20 kg [28,29,31]. Nevertheless, this technique has also been used in adult human patients for spinal fixation during spondylosis and spondylolisthesis [30]. Despite its limitations, this technique yielded good outcomes in our patients. Spinal stapling was performed in most cases of thoracolumbar SFL because of its cost-effectiveness, surgical ease and the need for small surgical incisions in wild animals. This choice was based also on the surgeon’s experience and confidence in performing spinal stapling in wild animals, an implant easy to remove with minimum tissue dissection in case of complications. In the wolf with an L3 fracture, the rotational instability of the vertebral body was considered excessive for spinal stapling, and a screw–PMMA combination was chosen. The use of screws or pins secured with PMMA has been described for stabilization of SFL in all locations with a relatively large vertebral body [7], and this technique is currently one of the most popular for spinal stabilization in dogs [8,29,32]. This procedure requires minimal instrumentation and is compatible with decompressive techniques but requires an excellent knowledge of vertebral anatomy to ensure accurate implant positioning [29]. Both pins and screws have been used in association with PMMA, and even though pins have a better bending strength according to Garcia et al. (1994) [33], they may have several disadvantages. Pins are easy to apply but, in comparison with cortical bone screws, are more likely to migrate, are less resistant to pullout [34] and, unlike screws, their strength is strongly influenced by their diameter [35]. Potential side effects of screw/pin–PMMA implants are risk of thermal injury caused by the exothermic reaction during the application of PMMA, difficult wound closure over bulking PMMA and a potential risk of infections. The incorporation of antibiotics into the polymer powder before mixing can reduce this risk [35]. Current treatment recommendations for cervical spinal injuries in dogs include bandages and splints, unless the neurological grade deteriorates [36]; however, there is no clear consensus on the best therapeutic approach for humans and animals [37]. Surgical treatment of cervical fractures in dogs is associated with high perioperative mortality, although functional recovery in perioperative survivors is excellent [38]. Regarding C2 fractures in dogs, Schmidli et al. (2019) found that the chances of complete or functional recovery were high for both conservative and surgical management. However, patients with a dislocation between C1 and C2 greater than 30% and patients with more severe signs were more likely to be treated surgically. Furthermore, the most commonly applied technique was a screw–PMMA combination [39]. In our patient with a cervical fracture, we opted for surgical treatment because of the characteristics of the fracture (type IIIa according to the Anderson and D’Alonzo modified axis fracture classification system, with a dislocation between C1 and C2 greater than 30% [39]), the neurological status (grade 3) and the challenge of an effective and prolonged neck splinting in a wild animal. External immobilization using neck splints may be adequate for humans with stable cervical fractures. However, in small animal patients, this treatment can be considered more difficult and less effective due to poor patient compliance, the absence of a collarbone and the inability of adequate immobilization because of excessive soft-tissue interposition [39]. Further, performing this procedure in wild animals is even more challenging because the need for the patient’s head and neck restraint and manipulation during routine bandage monitoring can put at risk the safety of both patients and operators. In view of all these considerations, a screw–PMMA implant combination was used in this animal. No implant was removed from the wolves before release into the wild. Implants may be removed in case of complications. Implant failure, infection, granuloma formation, gross malalignment, chronic pain caused by subclinical implant loosening and evidence of pressure necrosis which may lead to future implant migration are few examples. Except for external skeletal vertebral fixation, implants generally remain in site if they are not the cause of complication. Nevertheless, if necessary, they can be removed when stability is considerate appropriate [31,40] (Figure 3). This makes the use of spinal stapling, a limited invasive surgical approach, a favorable technique. Antimicrobial prophylaxis is usually considered sufficient to limit surgical site infection rate in spinal surgeries [41], which are usually clean procedures and rarely involve exposed fractures or contaminated wounds. Nevertheless, all our patients received a post-operative antibiotic course of a broad-spectrum antibiotic for two reasons. First, most patients presented with multiple contaminated skin lesions caused by the RTA. Second, there are inevitable limitations posed by the post-operative care and monitoring of wild animals. Specifically, all wolves were hospitalized during recovery at a rehabilitation center. Nonetheless, routine physical examinations and wound monitoring were kept to a minimum to minimize stress, potentially making subclinical wound infections more likely to go undetected. Moreover, even if the wolves were housed in clean facilities, these housing conditions were less aseptic than dogs’ normal post-operative shelter conditions. The use of corticosteroids in spinal surgery in both humans and small animals **is** controversial. One of the major concerns about the use of steroids during or after spinal surgery is the risk of developing gastrointestinal side-effects [42] or the increased risk of postoperative infections, especially urinary tract or wound infections [43]. However, the association between steroids and infections is controversial, with studies both supporting or questioning the association between them [44,45]. For instance, both Gardiner (2020) and Fletcher (2020) have shown that a short course of post-operative steroids dramatically decreases total opioid use and improves global pain control in people undergoing spinal surgeries, without affecting surgical outcome, postoperative wound-healing complications or the risk of acute infections. Despite this controversy surrounding steroids, the authors believe that they can still be used judiciously in appropriate situations [46,47]. Given their anti-inflammatory and membrane stabilization properties, steroids can be considered in surgeries involving significant spinal cord manipulation, such as SFL. A possible key to safely use these drugs is using them at an anti-inflammatory dose for a short tapering period. Spinal stabilization and rehabilitation of the wolves included in this study were performed to increase their chance of release into the wild. According to Art. 26 comma 6 bis and 62 comma 1 of the regional law 8/94, the ultimate aim of the C.R.A.S., after initial rescue, treatment and temporary detention for rehabilitation, is reintroduction in nature. Six wolves presented with a neurological grade of 0 and lesions compatible with a poor prognosis for neurological recovery. According to the wildlife health guidelines cited and Article 2 of the Italian law 157/92, if a rescued wild animal has injuries with a poor prognosis and its condition is associated with physical and unmanageable suffering, humane euthanasia is warranted. Euthanasia is also suggested if the animal can survive in captivity but has physical handicaps that dramatically reduce the quality of life. Therefore, we opted for euthanasia in all the paraplegic wolves without DPP and with poor prognosis. Furthermore, only animals with clinical, functional and neurological recovery (grade 6) were released into the wild. The wolves had no residual neurological deficits and could walk, run, hunt and manifest all the normal behaviors of a wild animal. Only one wolf was not released because it reached only grade 4 and was classified as “incurable” by Italian C.R.A.S. guidelines. This animal is permanently housed at the rehabilitation center for educational purposes. This study has limitations. First, the retrospective design prevented the inclusion of a control group of surgically or medically treated wolves with grade 0 to compare with our population, as paraplegic animals without DPP were always euthanized for ethical reasons. Second, no patient presented with grade 1 or 2 on the modified Frankel Scale. Third, our study population was small and included patients with fractures or luxations in different spinal segments. Nevertheless, to the best of our knowledge, our sample size was larger than any other previous neurosurgical study involving rescued wolves. Follow-up information was available for all cases but varied greatly. The follow-up of wildlife animals in rehabilitation centers is often limited by the duration of rehabilitation, as animals are released from captivity as soon as they are fit for survival to reduce stress and human influence. However, relevant information can still be obtained using telemetry collars, which were applied to all freed wolves.

## 5. Conclusions

To the best of our knowledge, this study is the first to describe traumatic spinal injuries in Italian wolves. Given the lack of scientific literature on SFL in this species, our decisions were made according to medical guidelines for the nearest domestic species, *Canis familiaris*. Our results showed that wolves with grade 3 or higher on the modified Frankel classification had good prognosis for functional recovery after either surgical or conservative treatment. None of the vertebral implants were removed, thus avoiding any long-term complications. Five out of six of the wolves treated surgically and one of the two wolves treated conservatively were released into the wild. Thus, wolves with a neurological grade 3 at presentation caused by SFL may achieve neurological and motor-function recovery after either surgical or conservative treatment, allowing release of these animals into the wild.

## Figures and Tables

**Figure 1 animals-12-03044-f001:**
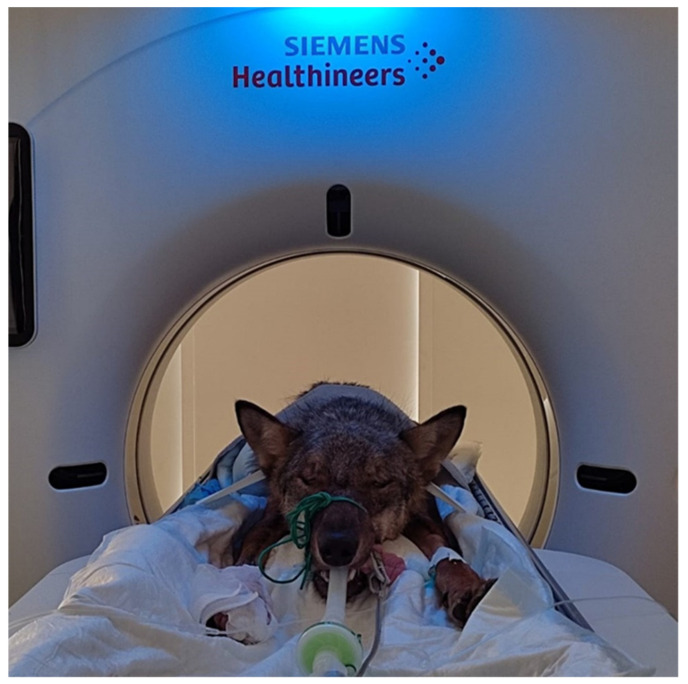
An Italian wolf during a CT exam.

**Figure 2 animals-12-03044-f002:**
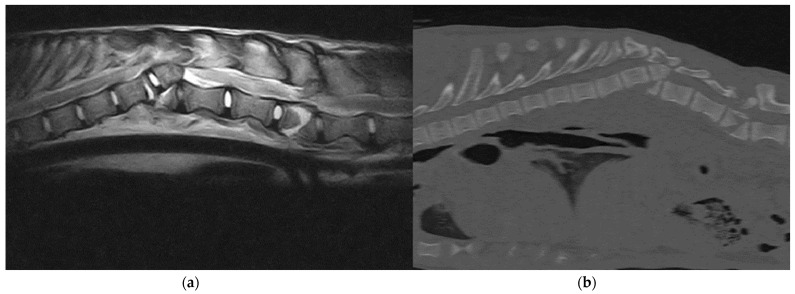
(**a**) Sagittal T2-weighted MRI and (**b**) sagittal bone window CT scan: A 6-month-old, female, Italian wolf affected by vertebral fracture and luxation at the level T12 and L2.

**Figure 3 animals-12-03044-f003:**
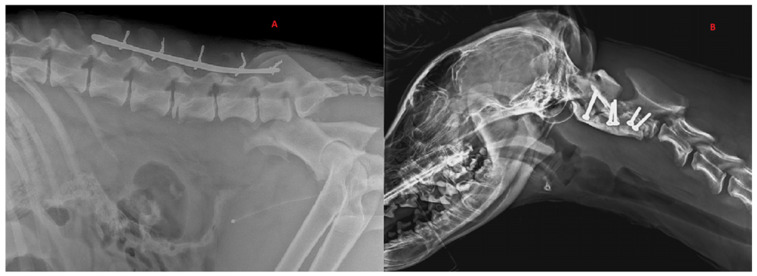
Radiographic exams. (**A**) Application of dorsal spinal staple in 50-month-old, male, Italian wolf with L5 vertebral fracture. Bilateral ileal: body oblique fractures; (**B**) application of ventral implant with screw–PMMA in 7-month-old, female, Italian wolf with C2 vertebral fracture and luxation.

**Table 1 animals-12-03044-t001:** Clinicopathological features in 14 Italian wolves.

Case No	Gender	Age (months)	Weight (Kg)	Cause	Vertebral Site	Other Injuries	Neurological Grade	Treatment	Final Outcome
1	M	23	34	HBC	L6	Hip Fracture	Grade 0	None	Euthanasia
2	M	28	31	HBC	T12, T13	None	Grade 0	None	Euthanasia
3	F	21	28	Discospondylitis	T12-T13	None	Grade 4	Body-splint and antibiotics treatment	10 weeks: Normal, Release into the wild (grade 6).
4	F	6	13	HBC	T12, L2	None	Grade 0	None	Euthanasia
5	M	15	27	HBC	L2	None	Grade 0	None	Euthanasia
6	M	10	20	HBC	T13	None	Grade 0	None	Euthanasia
7	M	12	28	HBC	L3	Hip fracture	Grade 3	Screws-PMMA implant	12 weeks: Normal, Release into the wild (grade 6).
8	F	7	11	HBC	C2	Humerus fracture Radius/ulna fracture	Grade 3	Screws-PMMA implant	15 weeks: Normal, Release into the wild (grade 6).
9	M	8	13	HBC	T11	None	Grade 4	Dorsal spinal staple	7 weeks: Normal, Release in the wild (grade 6).
10	M	7	16	Unknown	T5, T6, T7	None	Grade 3	Body-splint	3 weeks: Euthanized for poor quality of life
11	M	50	30	HBC	L5	Hip fracture	Grade 3	Dorsal spinal staple	16 weeks: Normal, Release into the wild (grade 6).
12	M	34	29	HBC	L2	None	Grade 3	Dorsal spinal staple	At present ambulatory with spinal ataxia. Stay in protection center (grade 4).
13	F	4	8	HBC	L2, L3	Ribs	Grade 0	None	Euthanasia
14	F	26	26	HBC	L2	Radius/ulna fracture	Grade 4	Dorsal spinal staple	6 weeks: Normal, Release into the wild (grade 6).

## Data Availability

The data presented in this study are available on request from the corresponding author.

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
