# Peer review of "Diagnostic and Treatment of Spinal Fracture and Luxation in Italian Wolves (Canis lupus italicus)"

_animals, 2022, doi:10.3390/ani12213044_

Round 1
Reviewer 1 Report
The manuscript “Diagnostic and treatment of spinal fracture and luxation in Italian wolves (Canis lupus italicus) in comparison with spinal injuries in dogs” describes the findings of clinical examination, diagnostic imaging and surgical treatment in 14 Italian wolves suffering spinal fractures secondary to presumptive or witnessed trauma.
Unfortunately, due to several major limitations, I cannot recommend this MS for publication in the current form. More work might be needed to fill important gaps:
-a more accurate analysis of similarities and differences in the clinical and radiological findings and management of spinal fractures in wolves vs dogs is expected (as claimed in the title.)
- confusing statements, particularly concerning applicability of clinical disability scales, prognostic factors, clinical course and outcome of spinal cord injury in dogs to wild animals, should be avoided/ammended
- the manuscript would profit from a thorough discussion on ethical, (animal) welfare and conservation aspects associated with hospitalization/management of spinal fractures in wolves, as these aspects have certainly a major impact on treatment choice and/or euthanasia in these animals vs pets
- in several parts, the MS appears carelessly prepared, overall, the MS could use a thorough job of editing and proof-reading to clarify many confusing statements.
Additional comments
Abstract
Line 13: [..] made us take into consideration the literature [..] please consider more appropriate word choice and style
Line 23: [..] Our results suggest that wolves that retain pain sensation prior to surgery have a good prognosis for functional recovery.[..] as wolves presented with no nociception were systematically euthanized (and it is not known if they could had regained ability to walk) this conclusion it is not fully supported.
Matherials and Methods
Line 44: According to the Italian law 152/1992, the wildlife protection centers (CRAS) can contact veterinary facilities that have an agreement for the care and welfare of wild animals [..] please consider revise the entire sentence and provide clearer and detailed information on current guidelines and ethics- understanding principle underlying ethical and welfare decision on such endangered species is probably the most important aspect of this MS as surgical and conservative treatment on spinal fractures in carnivores is described elsewhere.
Line 47: clinical examination: how was this performed in a wild animal? Was chemical immobilization needed? Which drugs were used? How have handling and restraint impacted neurological assessment? How was nociception tested? Important information is missed here!
Line 56: [..] the etiology of SFL[..] aetiology or cause? Was this available for all rescued animals? Was this known or unknown trauma? Was the “HBC” accident witnessed or suspected? please amend or delete the sentence if no information was available.
Line 64: [..] with a veterinary MRI [..] please specify characteristic (high vs low field)
Line 68: treatment and follow up. Important information is lacking in this section or within the entire MS. It is unclear how the neurological assessment was performed, if this was done according to established protocols existing for wildlife and under recognized ethical and welfare principle (see comments at line 44).
This is particularly important for animals left with impaired neurological function (the one with spinal ataxia and the one
Line 70-74: please consider re-write the entire paragraph and amend for clarity.
Line 77-79: [..] this section may be divided by subheadings. It should provide a concise and precise 77 description of the experimental results, their interpretation, as well as the experimental conclusions that can be drawn [..] Text should be revised carefully for typos/ copied text etc etc before upload
Line 123-4: [..]and an anti-inflammatory course of prednisolone starting from 0.5 mg/kg s.i.d for the first 7 days tapering gradually over the next 7 days. What is the rationale for this treatment as Steroids in this species? No evidence exists of superiority of steroids treatment vs NSAIDs in spinal cord injury in dogs, is this different in wolves? Is this due to costs? Please clarify and expand in the discussion
Results
Table:
The case number 12 is reported to “[..] stay in the protection center” at 8 weeks. Is this table reporting only 8 weeks follow-up of the cases– outcome or final outcome? This is not clear and should be clearly stated in both the legenda and MS
Discussion:
Line 196: [..] This finding may represent the significant force needed to cause SFL of the inherently stable vertebral column [..] unclear – please rewrite, explain and contextualize this sentence
Line 197-199: [..] Like in a previous study [8], even in our cases no significant relationships were detected between the cause trauma and the severity of injury[..] as it is stated on several occasions in the MS that the cause of trauma for these animals in unknown I struggle to understand this statement. Please explain or delete the entire sentence.
Line 203-205: [..] the real degree of instability is often difficult to predict [13]. Therefore, if available, advanced second level diagnostic is strongly suggested to better characterize the lesions and set up an appropriate therapeutic planning[..] although this reviewer strongly agree with the use of CT and MRI, it is not clear from this discussion why (i.e. because of lesion visualization in additional spatial planes?) these studies are superior to radiographic ones and why CT scan was performed in addition to MRI scan (i.e. to investigate in very short time distribution of additional lesions within the body/spinal segments? To retrieve accurate measurements for surgical planning?)- also- what about costs? What is the rationale for using both MRI and CT scan when superiority of CT is stated by the Authors?
Line 214: The most used surgical technique to stabilize the vertebrae in the present study was spinal stapling [..] it is not clear why this information is again found here in the discussion as no statements on benefit or limitations or clinical implications of one vs another technique, can be found in addition to the information in the “materials and methods” section.
Line 230: [..] the impossibility of an effective and prolonged neck splinting in a wild animal.[..] why this is not feasible in wolves, but “bodice splint” (TL splinting) is? Please explain and clarify the statement
Line 232: [..]implants can be removed if necessary [..] no clear recommendation exists on this topic, however it is unlikely or rare that implant removal is necessary. When a spinal implant should be removed in dogs in Author’s opinion? What necessary means here? (i.e. iatrogenic infection, implant failure?)
Author Response
Dear Reviewer 1,
please see the attachment

Reviewer 2 Report
Thank you for this interesting case series reporting the management of vertebral fracture/luxations in wolves. The manuscript is well organized and globally well written although I would recommend using a professional English editing service. Your choice to euthanatize wolves without DPP, even if easily understandable, must be better explained and justified based on recent literature. I would also recommend expanding a little bit on the use of vertebral stapling, emphasizing on the comparison with other means of stabilization (ie biomechanical, success rates in dogs,...)
Title:
To my opinion, "in comparison with spinal injuries in dogs" should not be placed in the title as the comparison is mainly for the purpose of the discussion. I suggest: "Management of spinal fractures and luxations in Italian wolves (Canis lupus italicus)"
Abstract:
Line 16: all data were
Line 17: I would suggest "road traffic accident"
Line 18: "small animals spine"? don't you refer to your cases there? so you mean "of vertebral fractures or luxations in Italian wolves."
Line 19: but you describe a 6-grades scale later in your manuscript. Please be consistent.
Line 22: deterioration of the neurological status
Line 23: I would add a small sentence mentioning the good recovery and release in wild
Line 23: with a retained
Introduction:
Line 31: prefer "road traffic accident"
Line 34: delete "first"
Line 37: steroid administration? steroid administration is of real benefit if performed within a very short time after the trauma. After few hours, there is no difference between NSAIDS and steroids. To my opinion, it would better to mention "analgesia and anti-inflammatory medication".
Line 37: "Surgical treatment involves stabilization of the affected vertebral segment by the means of ..."
Line 39: "or external skeletal fixation"
Line 40: "Canis lupus italicus"
Materials and Methods:
Line 48: delete "(Canis lupus italicus)" as it has already been stated before
Line 49: what do you mean by a "specialist consultation"?
Line 56: data collected comprised
Line 57: diagnostic imaging modality or technique?
Line 58: site of spinal fracture or luxation
Line 60: Did you perform any other tests while admitting the patients? Blood works? thoracic X-rays? blood pressure? I understand that you aim at focusing on the description of SFL but as you describe the management, you should, if performed, expand more on what has been performed. If not, maybe explain why in the discussion.
Line 70: treatment was elected
Line 73: data were collected
Results:
Line 77-79: delete these sentences
Line 81: delete "(Canis lupus italicus)"
Line 84: secondary to severe discospondyltis
Line 85: due
Line 86: 11 cases
Line 87: 2 cases
Line 90: 6 patients
Line 91: 3 patients
Line 91: 5 patients
Line 93: references for radiography, CT and MR can be deleted as already stated before.
Line 94: replace "advanced second level diagnostics" by "three-dimensional imaging techniques"
Line 96: well this is debatable and depends more on what you were looking for. For SFL detection and characterization, CT scan is superior to MRI. However, to further characterize the status of the spinal cord, MRI is superior to CT. I suggest to rephrase this part of the section in that way.
Line 98: Does that mean that you did not perform DV or VD projections? Indeed, you could have missed some SFL. The question is then: were these xrays really necessary as you had already planned to perform 3D imaging? Was it a manner to better document the cases?
Line 99: 13 wolves
Line 103: this should not be stated in the results but in the discussion. By the way, could you develop this aspect with references and numbers based on dog literature so that the reader is able to understand your decisions ?
Line 104: delete "(7.1%)"
Line 105: idem
Line 106: please rephrase this sentence. I am not able to clearly understand what was the etiology of the scoliosis.
Line 107: conservatively instead of medically
Line 110: was subsequently
Line 111: of clinical signs and quality of life
Lines 113-114: delete the descriptions of grades here and use Grade 3 and grade 4. "4 of 6 wolves". "2 wolves".
Line 115: "of 6 cases"
Line 117: spinal stapling
Line 118: only one screw? which implant? you mean that you used PMMA to secure the screws don't you?
Line 121: why did you recommend antibiotic therapy for all cases? as it is a clean procedure, could you not have avoided that?
Line 129: 5 out of 6
Line 131: released
Line 146: Radiographic exams.
As pelvis fractures are apparent on this radiographs, you have to describe them.
Line 146-147: delete canis lupus italicus
Line 191: of the 14 wolves presenting with SFL
You do not talk about the other fractures (long bones, pelvis). To my opinion, you should say what is been performed otherwise the reader could think that you only performed vertebral stabilization.
Discussion:
Line 194: please rephrase this sentence
Line 196: "this finding..." how ? I don't understand what you mean.
Lines 197-199: "Like in a previous... of complications." this statement has not been developed in the results. This information, somewhat interesting, cannot be dropped here in the discussion without having been developed earlier. Did you make stats to say that?
Line 199: site of the lesion
Line 201: three-dimensional imaging techniques
Lines 202-208: the information is important but this paragraph needs to be edited. Please reformulate.
Lines 211-213: Although I agree that patients without DPP have a poorer prognosis, this reference is very old and several papers have been published more recently emphasizing the fact that this prognosis was not as bad as reported before. Some factors seem important to consider when talking about the recovery such as the DPP loss for more than 24h, etc... Please expand on this. Finally, you can justify your management by the fact that patients are wild animals, and so the care that could be given to a dog may be very difficult to obtain in a wolf, or something in this way. (Laitinen and Puerto, 2005; Takahashi et al., 2020; Olby et al., 2022; Loughin et al., 2005; Jeffery et al., 2016; Aikawa et al., 2012; Wang-Leandron et al., 2017). To my opinion, this part of the discussion is very important and must be developed accordingly with recent literature.
Line 216: 2 of 4
Line 217: 2 cases
Line 219: Please expand on a comparison of the techniques (biomechanical, results,...). This technique is clearly not recommended for patients of this weight and with severe displacements of the vertebrae. Maybe did you decide to use this technique in view of the very minor displacement of the vertebrae?? If yes, please state it, if not please justify more why you decided to use this technique, knowing the "inferiority" (at least for this type of patient).
Line 221: why did you decide to use screws and PMMA for the second patient?
Line 223: with purposeful?
Line 224: yes indeed but then why did you use surgical stapling for most of the cases? it is a bit dangerous for you to state that..
Lines 227-228: please rephrase
Line 229: because of the characteristics
Line 232: released
Conclusions:
Line 239: well, I don't agree as you decided to use vertebral stapling for patients of approximately 20kg. Please delete this sentence or reformulate.
Line 241: or higher
Line 247: released
Author Response
Dear Reviewer 2,
please see the attachment
Best regards

Round 2
Reviewer 2 Report
Thank you for having taking into account my suggestions. The manuscript is now substantially improved. The discussion, even if a little long with the many points suggested and added, is clear and addresses the important issues. There is still a work of editing on the punctuations, spaces between the words.
L13: based on
L15: between C1 and L7
L31: include
L31: due to animal attacks
L33: I suggest "Therapeutic management can be either conservative or surgical"
L49: by the IUCN in Italy
L52: As stated in my previous comments, what does "specialized examination" mean? Do you mean that you referred to a board-certified veterinary neurologist or surgeon? I would use this term with caution.
L58: I suggest: Medical records were reviewed and data regarding .... we're collected.
L64: on conscious animals
L76: workup exams were performed
L80: please delete this sentence and add the information to L58
L84: of these modalities.
L84: Radiographic
L87: suspected spinal injury
L97: spinal cord involvement should be reformulated and be more specific
L97: "in this paper...information". Please delete this sentence
L106: a range of dose should be added for propofol
L134: I suggest "All 14 animals underwent latero-lateral and dorso-ventral radiographic projections of the spine, thorax and abdomen under sedation."
L137: affected spinal segment
L148: The 6 paraplegic wolves
L161: was it always the same size for the Steinmann pin? can you add a range of size or precise if it was always the same?
L164: I suggest: Loops of cerclage wires were then threaded in each hole and tightened around the pin
L175: None of the patients
L256: I don't really like the term "spinal cord involvement". There is of course a spinal cord involvement otherwise the patient wouldn't be paraparetic/plegic. I would argue that this modality is able to better characterize intramedullary lesions and eventually give prognostic factors.
L278: prefer accurate to precise
L309: please provide the reference for this classification
L430: spinal segments
L439: describe
